# Anesthetic technique and postoperative pulmonary complications (PPC) after Video Assisted Thoracic (VATS) lobectomy: A retrospective observational cohort study

Priyanka Singla[1]*, Brian Brenner[1], Siny Tsang[1], Nabil Elkassabany[1], Linda W. Martin[2], Phillip Carrott[2], Christopher Scott[2], Michael Mazzeffi[1]

1 Department of Anesthesiology, University of Virginia, Charlottesville, Virginia, United States of America,
2 Department of Surgery, University of Virginia, Charlottesville, Virginia, United States of America

* ps7ey@uvahealth.org

## Abstract

### Introduction

Thoracic surgery is associated with an 8–10% incidence of postoperative pulmonary complications (PPCs). Introduction of minimally invasive Video-assisted thoracoscopic surgery (VATS) aimed to reduce pain related and pulmonary complications. However, PPCs remain a common cause of morbidity after VATS.

### Methods

The American College of Surgeons National Surgical Quality Improvement Program database was searched for VATS lobectomy cases from 2017 to 2021 with General Anesthesia (GA) as the primary anesthetic technique. Cases were stratified into four groups—GA alone, GA+local, GA+Regional and GA+Epidural. Generalized linear regression models were used to examine whether PPCs differ by anesthetic technique, controlling for morbidity risk factors. The study's primary outcome was the occurrence of any PPC (pneumonia, reintubation or prolonged mechanical ventilation). The secondary outcome was length of hospital stay (LOS).

### Results

A total of 15,084 VATS lobectomy cases were identified and 14,477 cases met inclusion criteria. The PPC rate was between 3.5–5.2%. There was no statistically significant difference in the odds of PPCs across the groups. Compared to the GA alone group, the regional and local group had significantly shorter LOS (9.1% and 5.5%, respectively, both ps < .001), whereas the epidural group had significantly longer LOS (18%, p < .001).

### Conclusion

Our analysis suggests that the addition of regional or local anesthesia is associated with shorter LOS after VATS lobectomy. However, these techniques were not associated with

**Data Availability Statement:** Data is available from ACS NSQIP database (https://www.facs.org/quality-programs/data-and-registries/acs-nsqip/

participant-use-data-file/). The ACS NSQIP database and the hospitals participating in ACS NSQIP are the source of the data used herein; they have not verified and are not responsible for the statistical validity of the data analysis or the conclusions derived by the authors.

**Funding:** The author(s) received no specific funding for this work.

**Competing interests:** The authors have declared that no competing interests exist.

lower PPC incidence. Future randomized controlled trials could help to elucidate the best anesthetic technique to reduce pain and enhance recovery.

## Introduction

Lung cancer is the third most common cause of cancer and leading cause of cancer death in the United States [1]. Video-assisted thoracoscopic surgery (VATS) lobectomy is one of the most common surgical procedures performed to treat lung cancer [1, 2].

Uncontrolled postoperative pain after thoracic surgery is a major cause of morbidity and poor outcomes including respiratory complications, prolonged hospital stay, poor quality of life, and chronic persistent postoperative pain [3]. The development of VATS led to significant clinical improvements including reduced surgical trauma as well as severity of acute and incidence of chronic pain, shorter hospital stay and decreased morbidity [4]. Patients' postoperative quality of life also improved compared to after open surgery [2]. Nevertheless, postoperative pulmonary complications (PPCs) remain an important and relatively common cause of morbidity after VATS [5].

Along with the development of minimally invasive surgical techniques, advances in perioperative care such as the development of Enhanced Recovery After Surgery pathways (ERAS) aim to reduce complications related to postoperative pain and excessive opioid use [6, 7]. While thoracic epidural analgesia (TEA) is the gold standard for pain control after thoracotomy, there is no current standard of care for postoperative analgesia after VATS [8, 9].

New ultrasound-guided nerve blocks have gained traction as alternatives to TEA for VATS patients. A recent meta-analysis suggests that regional anesthesia techniques performed for pain control after VATS are non-inferior to TEA for postoperative pain control [8]. While pain is often used as one of the surrogates for PPC risk, there is little evidence comparing the impact of various adjuvant analgesic techniques on PPCs after VATS lobectomy.

The aim of our study was to explore the relationship between various analgesic techniques and PPCs after VATS lobectomy using the American College of Surgeons (ACS) National Surgical Quality Improvement Program (NSQIP) database. We hypothesized that patients receiving adjuvant local anesthesia, regional anesthesia, and TEA would have lower odds of PPCs and shorter LOS compared to patients receiving general anesthesia (GA) alone.

## Methods

### Patients

The Institutional Review Board at the University of Virginia School of Medicine exempted the study and waived the requirement for written informed consent. The authors did not have access to information that could identify individual participants during or after data collection.

We performed a retrospective cohort study using data from ACS NSQIP Participant User Files (PUF) January 2017 to December 2021. ACS NSQIP collects over 150 variables including preoperative risk factors, intraoperative variables, and 30-day postoperative morbidity and mortality for patients undergoing major surgical procedures. 658 hospitals contributed data to ACS NSQIP in 2021 [10]. Data are abstracted by trained surgical case reviewers at participating hospitals and periodic audits are performed to ensure interrater reliability. Prior studies on anesthetic technique using NSQIP data have been published [11, 12].

VATS lobectomy cases were included in the analysis if they meet the following inclusion criteria: (i) elective surgery, (ii) GA was the primary anesthetic technique, and (iii) operative time was $\geq$ 30 minutes. Patients less than 18 years of age were excluded. A total of 14,477 cases met the inclusion criteria. Patients were stratified into four groups based on whether an additional analgesic technique was performed, GA alone ($n$ = 9,062), GA + local ($n$ = 1,714), GA + Regional ($n$ = 3,069), and GA + TEA ($n$ = 632) (Fig 1).

## Outcomes

Three postoperative respiratory complications (new onset post-operative pneumonia, reintubation, and mechanical ventilation > 48 hours, recorded for 30 days postoperatively) were used to create a composite dichotomous PPC outcome (0 = no PPC, 1 = any PPC), which was the study's primary outcome. Total length of hospital stay (LOS) in days was a secondary outcome.

## Statistical analysis

Statistical analysis was performed using R 4.1.2 [13] Descriptive statistics were summarized for patient and operative characteristics. Generalized linear regression models (GLMs) were utilized to explore differences in PPCs and LOS across the four anesthetic groups (reference: GA alone). A binomial link was used for the binary outcome (any post-operative PPC), whereas a Poisson link was used for the count outcome (LOS). Differences in NSQIP predicted morbidity and mortality across the four anesthetic groups were also explored using GLMs. Predicted morbidity and mortality were log-transformed to correct for skewness.

Covariates included in the models were based on published risk factors used by NSQIP to compute predicted postoperative morbidity and mortality. A total of 18 risk factors were included: age group (< 65yo, 65-74yo, 75-84yo, and 85+yo), sex (male/female), functional status (independent/partially dependent/totally dependent), ASA physical status [1–5], steroid use for chronic condition (yes/no), ascites within 30 days prior to surgery (yes/no), systemic sepsis within 48 hours prior to surgery (none, SIRS, sepsis, septic shock), ventilator dependent (yes/no), disseminated cancer (yes/no), diabetes (no, oral, insulin), hypertension (HTN) requiring medication (yes/no), Congestive Heart Failure (CHF) in 30 days prior to surgery (yes/no), dyspnea (no, with moderate exertion, at rest), current smoker within 1 year (yes/no), history of severe Chronic Obstructive Pulmonary Disease (COPD) (yes/no), dialysis (yes/no), acute renal failure (yes/no), and Body Mass Index (BMI). As emergency cases were excluded from our study, emergency status was not included as a covariate in our analyses. In addition, operative time (hours) and year of surgery were included as covariates. In all models, age was divided by 10 to allow variables to be on a similar scale. For the any PPC outcome, partially and totally dependent were aggregated into one category due to the small number of cases with totally dependent functional status (0.01%), and ventilator dependent was not included as only one ($n$ = 1) case was ventilator dependent. Cases with missing covariates were excluded from the models. Statistical significance level was set at $\alpha$ = .05.

## Results

There were 15,084 VATS lobectomy cases between 2017 and 2021 in the NSQIP PUFs. The final analytic sample consisted of 14,477 VATS cases (Fig 1). Of these, there were 9,062 cases performed with GA alone, 1,714 cases performed with GA + local, 3,069 cases performed with GA+ Regional, and 632 cases performed with GA + TEA. There was an increased use of GA + Regional over time; each additional year is associated with 21% increase in the number of GA + Regional cases (IRR = 1.21, 95% CI = 1.18–1.25, $p$ < .001). In fact, the proportion of

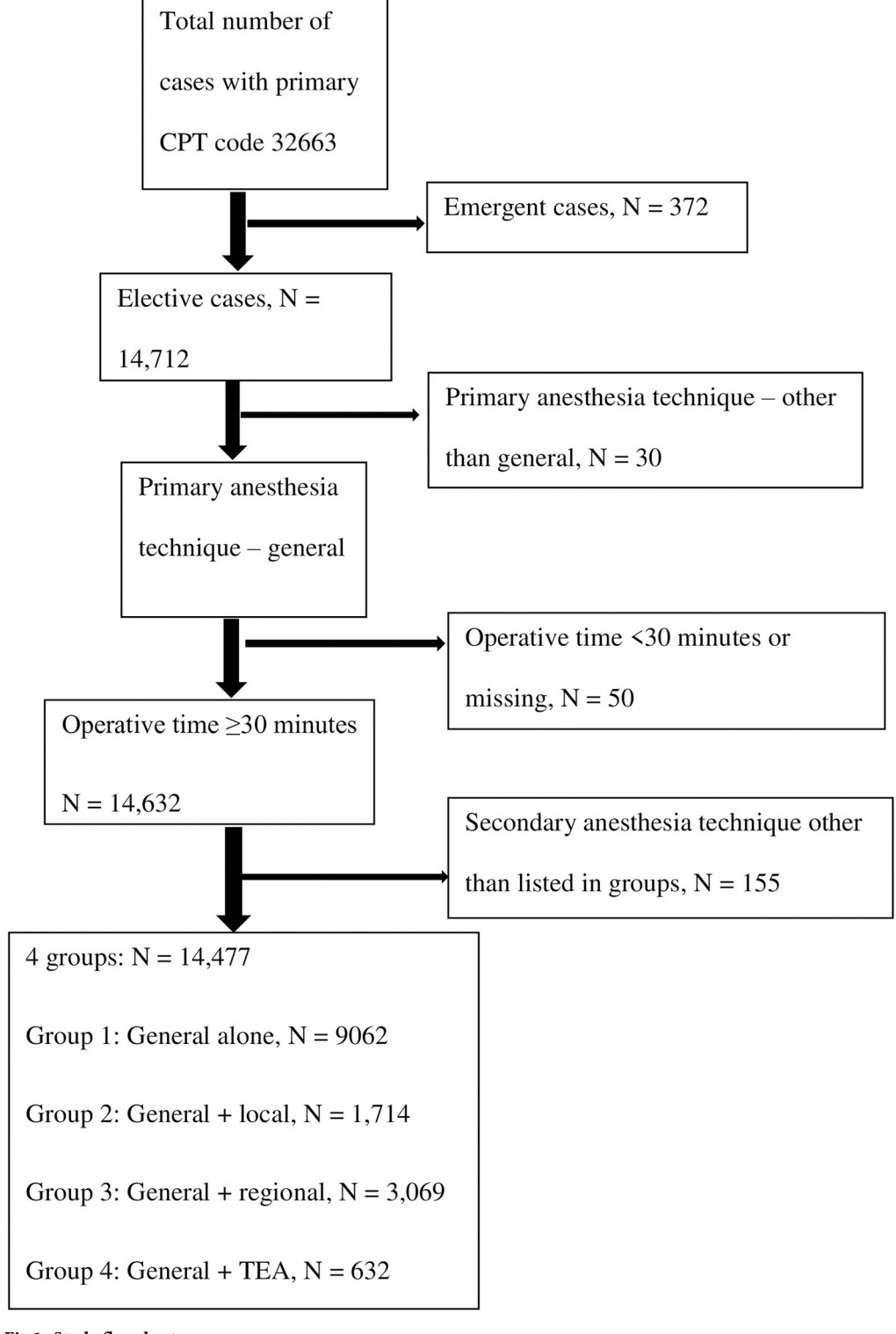

**Fig 1. Study flowchart.**

cases with use of adjunct regional anesthesia more than doubled over the five-year period from 12.7% in 2017 to 28.6% in 2021. On the other hand, utilization of TEA declined during the same time period from 5.9% in 2017 to 2.8% in 2021; each additional year is associated with 12% decrease in the number of cases utilizing TEA (IRR = 0.88, 95% CI = 0.83–0.93, $p < .001$) (S1 Table).

Demographic and clinical characteristics of cases that met the study inclusion criteria are shown in Table 1. S2 Table shows characteristics after stratification by group. Patients' age ranged from 18 to 90 years old (*Mean* = 67, *SD* = 10). The majority of patients (58.8%) were female. Most participants identified as White (68.0%) and non-Hispanic (94.9%). The percentage of ASA physical status score 4 patients was more than double in the TEA group (14.9%) compared to the GA alone (7%), GA + Regional (5%), and GA + local (6.2%) group (S3 Table).

## Postoperative pulmonary complications

As shown in Table 2, the incidences of any PPC event ranged from 3.5% (GA alone/GA + local) to 5.2% (GA + TEA). Results from logistic regression model showed no statistically significant difference in the odds of having a PPC when regional, local, and TEA groups were compared against the GA alone group (Table 3). Post-hoc analyses showed no statistically significant pair-wise differences. As illustrated in Fig 2 (left panel), the 95% CI of the odds ratio for the GA + regional, GA + local, and GA + TEA groups all crossed the OR = 1 threshold, reflecting the non-significant difference in the odds of PPC when compared against the GA alone group.

As shown in Table 3, morbidity risk factors associated with increased odds of any PPC included older age, male gender, increased functional dependence, elevated ASA score, use of steroids, HTN requiring medication, current smoker within 1 year, history of severe COPD, and increased operative duration.

## Length of stay

The median LOS ranged between 3 and 4 days (Table 2). Results from Poisson regression model showed statistically significant decrease in LOS for the GA + regional (IRR = 0.91, 95% CI = 0.87–0.94) and GA + local group (IRR = 0.95, 95% CI = 0.9–0.99), when compared to the GA alone group (Table 3). Compared with the GA alone group, LOS decreased by 9.1% and 5.5% for the GA + Regional and GA + local group, respectively. In contrast, there was a statistically significant increase in LOS for the GA + TEA group (IRR = 1.18, 95% CI = 1.1–1.26) when compared with the GA alone group, reflecting an 18% increase in LOS (Fig 2, right panel). Fig 3 illustrates predicted LOS based on anesthetic group. The difference in average predicted in LOS for local and regional group compared to GA alone group is however small (<1 day).

As shown in Table 3, morbidity risk factors linked to increased LOS included older, male gender, being partially dependent, elevated ASA score, use of steroids, sepsis, diabetes, HTN requiring medication, dyspnea, current smoker within 1 year, history of severe COPD, dialysis, decreased BMI, increased operative duration, and earlier surgery year.

Results from linear regression models showed no statistically significant difference in the probability of morbidity/mortality between the GA + Regional and GA alone groups. The probability of morbidity in the GA + local group was slightly higher (~2%), though the probability of mortality was not statistically significantly different, when compared to the GA alone group. Compared to the GA alone group, cases in the GA + TEA group had significantly higher probability of morbidity (~5.9%) and mortality (~12.2%) (S4 Table).

**Table 1. Descriptive statistics of patient demographics and clinical characteristics (N = 14477).**

| | | n | % |
|---|---|---|---|
| Age | < 65yo | 5158 | 35.6 |
| | 65 – 74yo | 5243 | 36.2 |
| | 75 – 84yo | 3841 | 26.5 |
| | 85yo+ | 236 | 1.6 |
| Sex | Male | 5967 | 41.2 |
| | Female | 8510 | 58.8 |
| Race[1] | White | 9817 | 68 |
| | Black/African American | 874 | 6 |
| | Asian | 575 | 4 |
| | Other | 3211 | 22 |
| Ethnicity[1] | Hispanic | 591 | 5.1 |
| Functional health status[2] | Independent | 14353 | 99.3 |
| | Partially dependent | 103 | 0.7 |
| | Totally dependent | 2 | 0.0 |
| ASA class[3] | 1 = No disturb | 30 | 0.2 |
| | 2 = Mild disturb | 2797 | 19.3 |
| | 3 = Severe disturb | 10635 | 73.5 |
| | 4 = Life threat | 985 | 6.8 |
| | 5 = Moribund | 1 | 0.01 |
| Steroid use for chronic condition | No | 13946 | 96.3 |
| | Yes | 631 | 3.7 |
| Ascites within 30 days prior to surgery | No | 14474 | 100 |
| | Yes | 3 | <0.1 |
| Systemic sepsis within 48 hours prior to surgery | None | 14404 | 99.5 |
| | SIRS | 66 | 0.5 |
| | Sepsis | 6 | <0.1 |
| | Septic shock | 1 | <0.1 |
| Ventilator dependent | No | 14476 | 100 |
| | Yes | 1 | <0.1 |
| Disseminated cancer | No | 13659 | 94.5 |
| | Yes | 818 | 5.7 |
| Diabetes | No | 12052 | 83.2 |
| | Non-insulin | 1743 | 12.0 |
| | Insulin | 682 | 4.7 |
| HTN requiring medication | No | 6150 | 42.5 |
| | Yes | 8327 | 57.5 |
| CHF in 30 days prior to surgery | No | 14381 | 99.3 |
| | Yes | 96 | 0.7 |
| Dyspnea[4] | No | 9630 | 82.9 |
| | Moderate exertion | 1925 | 16.6 |
| | At rest | 61 | 0.5 |
| Current smoker within 1 year | No | 9919 | 68.5 |
| | Yes | 4558 | 31.5 |
| History of severe COPD | No | 11616 | 80.2 |
| | Yes | 2861 | 19.8 |
| Dialysis | No | 14430 | 99.7 |
| | Yes | 47 | 0.3 |

(*Continued*)

**Table 1.** (Continued)

|  |  | *n* | % |
|---|---|---|---|
| Acute renal failure[5] | No | 11613 | 100 |
|  | Yes | 3 | <0.1 |
| Year of operation | 2017 | 2672 | 18 |
|  | 2018 | 2970 | 21 |
|  | 2019 | 3078 | 21 |
|  | 2020 | 2896 | 20 |
|  | 2021 | 2861 | 20 |
|  |  | *M* | *SD* |
| BMI |  | 28.2 | 6.0 |
| Operation time (min)[1] |  | 180.5 | 80.0 |

*M* = mean. *SD* = standard deviation.

[1]Not one of the risk factors to compute morbidity

[2]Unknown/missing for *n* = 19.

[3]No ASA class assigned for *n* = 29.

[4]Missing for *n* = 2861.

[5]Missing for *n* = 2861.

ASA = American Society of Anesthesiologists, BMI = body mass index, CHF = congestive heart failure,

COPD = chronic obstructive pulmonary disease, HTN = hypertension

## Discussion

In our five-year retrospective cohort study of over 14,000 VATS cases, we found that the incidence of any PPCs was between 3.5 and 5.2%. This is much lower than 8 to 10% incidence [14, 15] of PPCs noted by other authors. Air leak is one of the common complications after VATS lobectomy, however, this was not included in our composite PPC outcome [14, 16]. This could possibly explain the lower incidence of PPCs in our study. In addition, our results showed no association between anesthetic technique and odds PPCs, after controlling for morbidity risk factors, operation time, and surgery year. However, there was a significant correlation between anesthetic technique and LOS; patients who received GA + Regional and GA+ local had

**Table 2. Descriptive statistics of postoperative pulmonary complications and length of stay by anesthetic group.**

|  | GA alone (*n* = 9062) | GA + regional (*n* = 3069) | GA + local (*n* = 1714) | GA + TEA (*n* = 632) |
|---|---|---|---|---|
|  | *N* (%) | *N* (%) | *N* (%) | *N* (%) |
| Post-op PPC |  |  |  |  |
| 0 | 8747 (96.5%) | 2924 (95.3%) | 1654 (96.5%) | 599 (94.8%) |
| 1 | 216 (2.4%) | 106 (3.5%) | 48 (2.8%) | 24 (3.8%) |
| 2 | 55 (0.6%) | 21 (0.7%) | 6 (0.4%) | 3 (0.5%) |
| 3 | 44 (0.5%) | 18 (0.6%) | 6 (0.4%) | 6 (0.9%) |
| Any post-op PPC | 315 (3.5%) | 145 (4.7%) | 60 (3.5%) | 33 (5.2%) |
|  | Median [IQR] | Median [IQR] | Median [IQR] | Median [IQR] |
| Length of stay (days) | 3 [2 – 5] | 3 [2 – 5] | 3 [2 – 5] | 4 [3 – 6] |

GA = general anesthesia, IQR = inter-quartile range, PPC = postoperative pulmonary complication

**Table 3. Regression results estimating association between anesthetic technique and post-operative pulmonary complications and length of stay.**

| | Any PPC | | | Length of stay | | |
|---|---|---|---|---|---|---|
| | OR | 95% CI | p | IRR | 95% CI | p |
| Intercept | 4.13 | 0–>100 | 0.988 | >100 | >100->100 | <0.001 |
| Group: Regional | 1.13 | 0.88–1.46 | 0.324 | 0.91 | 0.87–0.94 | <0.001 |
| Group: Local | 0.94 | 0.68–1.31 | 0.729 | 0.95 | 0.90–0.99 | <0.001 |
| Group: Epidural | 1.44 | 0.97–2.13 | 0.070 | 1.18 | 1.10–1.26 | <0.001 |
| Age: 65-74yo | 1.43 | 1.11–1.85 | 0.006 | 1.12 | 1.08–1.16 | <0.001 |
| Age: 75-84yo | 1.73 | 1.32–2.28 | <0.001 | 1.18 | 1.14–1.23 | <0.001 |
| Age: 85yo+ | 2.13 | 1.07–4.25 | 0.032 | 1.41 | 1.27–1.57 | <0.001 |
| Sex: Male | 1.23 | 1.01–1.50 | 0.042 | 1.05 | 1.02–1.08 | <0.001 |
| Function status: Partially dependent[1] | 2.47 | 1.19–5.16 | 0.016 | 1.28 | 1.09–1.50 | <0.001 |
| Function status: Totally dependent | - | - | - | 1.13 | 1.04–1.23 | 0.767 |
| ASA | 1.56 | 1.26–1.94 | <0.001 | 1.09 | 1.06–1.12 | <0.001 |
| Steroid: Yes | 1.73 | 1.14–2.61 | 0.009 | 1.13 | 1.03–1.24 | <0.001 |
| Ascites: Yes | 0 | 0–>100 | 0.983 | 0.43 | 0.34–0.56 | 0.152 |
| Sepsis: SIRS | 0.74 | 0.17–3.21 | 0.687 | 1.19 | 0.97–1.46 | 0.006 |
| Sepsis: Sepsis | 0 | 0–>100 | 0.984 | 4.64 | 2.19–9.86 | <0.001 |
| Ventilator dependent: Yes[2] | - | - | - | 2.26 | 1.60–3.18 | 0.279 |
| Disseminated cancer: Yes | 0.66 | 0.37–1.16 | 0.151 | 0.98 | 0.92–1.05 | 0.423 |
| Diabetes: Non-insulin | 0.79 | 0.57–1.08 | 0.143 | 0.96 | 0.92–1.01 | 0.013 |
| Diabetes: Insulin | 1.19 | 0.78–1.81 | 0.413 | 1.10 | 1.01–1.20 | <0.001 |
| HTN requiring medication: Yes | 1.25 | 1.00–1.56 | 0.046 | 1.02 | 0.99–1.05 | 0.047 |
| CHF in 30 days prior to surgery: Yes | 0.93 | 0.27–3.18 | 0.902 | 1.06 | 0.86–1.31 | 0.418 |
| Dyspnea: Moderate exertion | 1.20 | 0.95 0 1.52 | 0.129 | 1.11 | 1.06–1.15 | <0.001 |
| Dyspnea: At rest | 1.35 | 0.51–3.56 | 0.547 | 1.12 | 0.90–1.40 | 0.043 |
| Current smoker within 1 year: Yes | 1.55 | 1.25–1.92 | <0.001 | 1.12 | 1.08–1.16 | <0.001 |
| Hx COPD | 2.28 | 1.84–2.83 | <0.001 | 1.22 | 1.17–1.27 | <0.001 |
| Dialysis: Yes | 0.99 | 0.23–4.34 | 0.988 | 1.16 | 0.89–1.52 | 0.041 |
| Acute renal failure: Yes | 0 | 0–>100 | 0.982 | 1.40 | 1.22–1.62 | 0.240 |
| BMI | 0.91 | 0.76–1.08 | 0.281 | 0.90 | 0.88–0.93 | <0.001 |
| Operation time (hours) | 1.25 | 1.17–1.33 | <0.001 | 1.10 | 1.08–1.11 | <0.001 |
| Surgery year | 1 | 0.91–1.09 | 0.936 | 0.98 | 0.97–1.00 | <0.001 |

OR = odds ratio. IRR = incident rate ratio. 95% CI = 95% confidence intervals. BMI = body mass index. Hx COPD = history of severe COPD, PPC = postoperative pulmonary complication

[1] Partially dependent and totally dependent were combined due to limited *N* in the any post-op complications model.

[2] Not included in the any post-op complication model as only one case (*n* = 1) was ventilator dependent.

shorter LOS whereas patients who had GA + TEA had longer LOS, when compared to GA alone.

Inadequate postoperative analgesia after lung resection is associated with atelectasis, reduced cough and difficulty clearing secretions, which increases the risk for pneumonia and acute respiratory failure after surgery [14, 16, 17]. In our study, we found no statistically significant association between anesthetic technique and PPCs. It is possible that the different anesthetic techniques analyzed led to similar pain scores for minimally invasive surgery. However, we are not able to analyze pain scores in this dataset. Multiple authors have found similar results when comparing various analgesic techniques [17, 18]. For example, in a retrospective analysis of 62 patients, Haager et al. found no difference in respiratory complications between

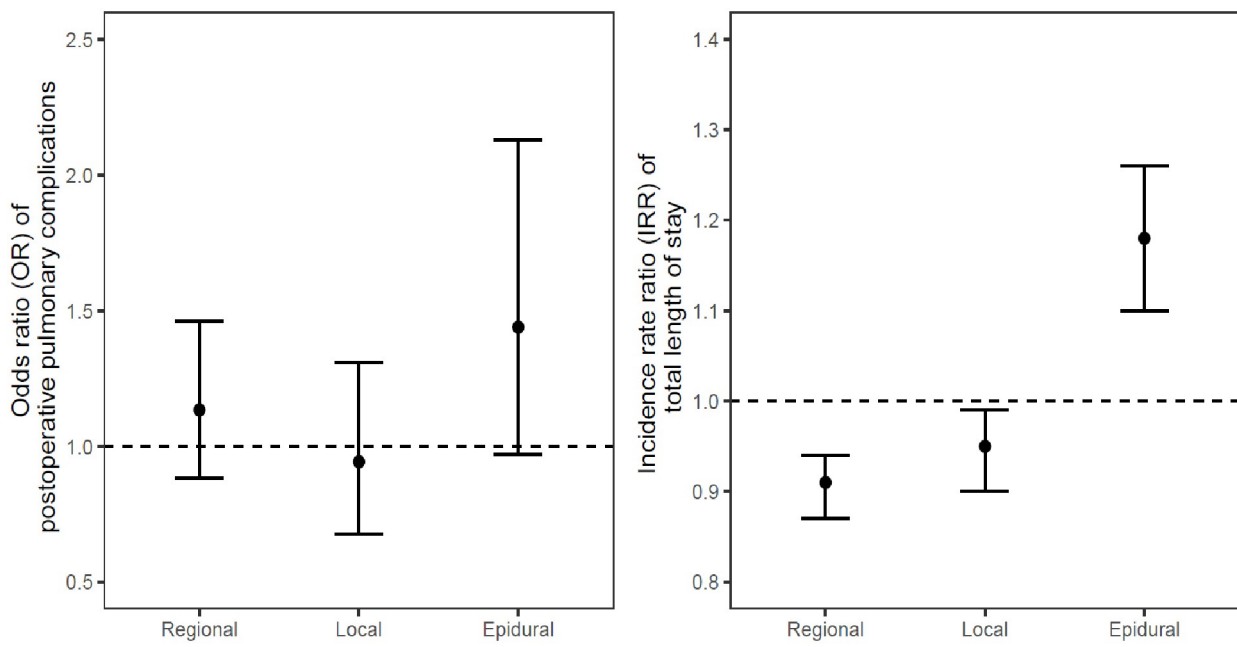

**Fig 2.** Odds ratio of PPCs comparing anesthesia techniques against the General group (OR = 1 refers to equal odds of PPC between the GA group and the comparison groups)–left panel; Incidence rate ratios of total length of stay (LOS) comparing anesthesia techniques against General group —right panel. Estimated ORs/IRRs (95% CI) are denoted with solid circles (whiskers). The dashed line represents OR/IRR = 1 where there is no difference in odds ratio or incidence rate ratio between GA-only and the comparison groups (GA + Regional, GA + local, and GA + TEA).

patients receiving TEA, thoracic paravertebral blocks, and systemic opioids [17]. In a recent meta-analysis that compared GA + TEA vs. GA alone for thoracic and upper abdominal surgery, the authors concluded that the benefits of TEA were ambiguous and, if present, they decreased over time [19].https://paperpile.com/c/gLaWGn/mBIc

During the five-year study period, the proportion of cases utilizing regional techniques increased, while the percentage of patients receiving TEA decreased from 24.8% of cases in the dataset in 2017 to 12.7% in 2021. (S1 Table) This could be attributed to introduction of various ultrasound-based techniques such as the Erector Spinae Plane block (ESP) and Serratus Anterior Plane block (SAP), which are less invasive and have lower complication, such as hypotension, urinary retention as compared to TEA [20–22]. Regional anesthesia techniques and local infiltration do not cause a sympathectomy and so there may be less risk for hypotension. Conversely, the risk of postoperative cardiac events is probably not reduced by these blocks; whereas TEA may reduce postoperative cardiac complications after thoracic surgery [23]. Multiple studies have shown the benefit of using regional blocks to improve postoperative pain scores and decrease systemic opioid use [20–22].

Long-acting local anesthetic (e.g. liposomal bupivacaine) infiltration at the operative site has been studied extensively in thoracic surgery and has been shown to improve pain scores, reduce opioids use, and reduce length of stay [24–26]. Benefits of long-acting local anesthetic infiltration in comparison to TEA include ease of performance, no risk for epidural hematoma, and no need for an indwelling invasive catheter. The potential duration of analgesia is up to 72 hours with liposomal bupivacaine.

Reasons for delayed discharge after VATS lobectomy include air leak, pneumonia and uncontrolled pain [16]. The main benefit of adding regional anesthesia or local infiltration based on our study results appears to be shorter LOS. This is consistent with prior studies [26–

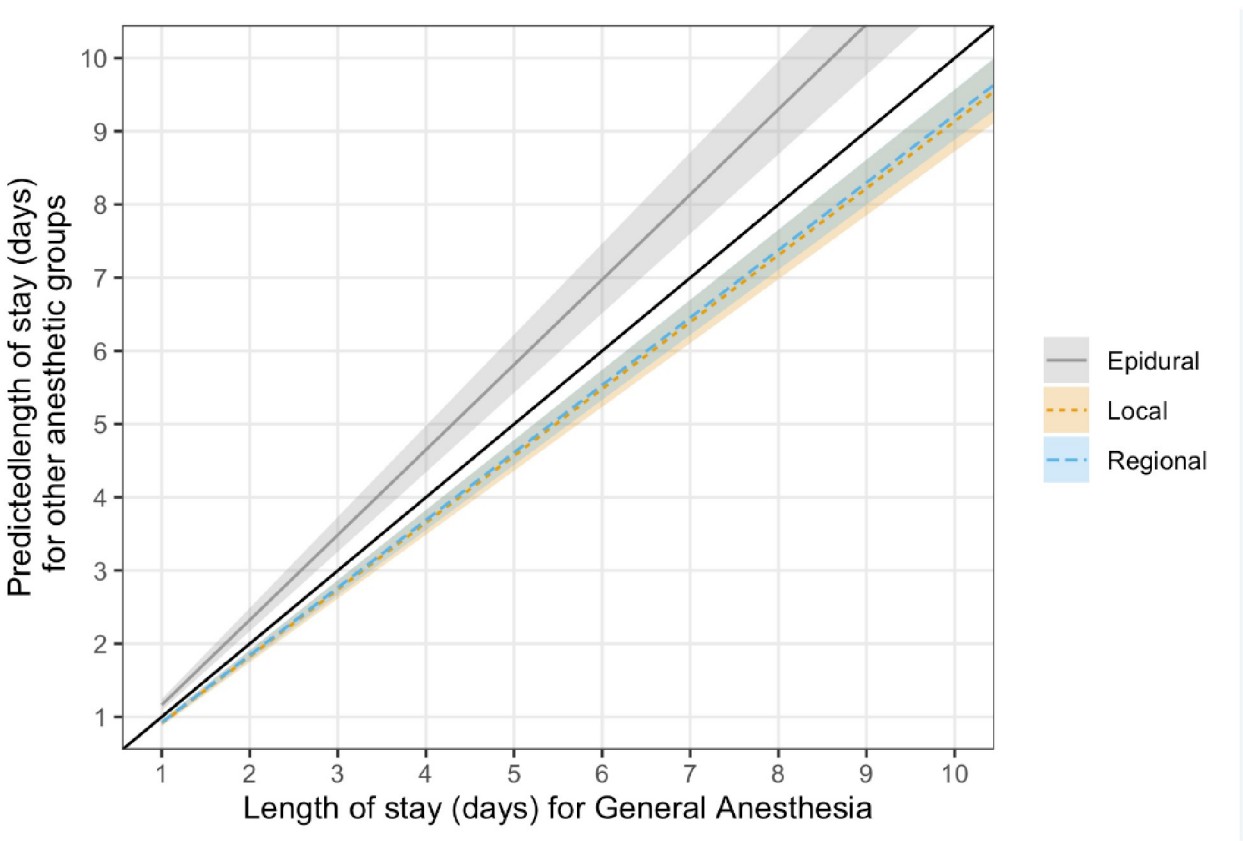

**Fig 3. Predicted length of stay based on anesthetic group.** The x-axis represents the length of stay (in days) for patients who received GA, whereas the y-axis represents the predicted length of stay for patients who received epidural, local, or regional anesthesia. The black reference line reflects the predicted LOS for epidural, local, and regional anesthesia if there was no difference in LOS as compared to the GA group. The average predicted LOS for the epidural group (grey solid line) is above that of the reference line (black solid line), reflecting that patients who received epidural are more likely to have longer LOS as compared to those who received GA. For example, a patient who received GA may have 3 days LOS, whereas the same patient may have >3 days LOS if they had received epidural. This difference is magnified with increased LOS. On the other hand, the average predicted LOS for the local (tan dotted line) and regional (blue dashed line) is below that of the reference line (black solid line), suggesting that patients who received local or regional are likely to have shorter LOS as compared to those who received GA. Of note, the difference in LOS is relatively small (< 1 day difference).

28]. Both analgesic techniques reduce opioid consumption, which probably reduces gastrointestinal complications (e.g. ileus nausea, vomiting) and other opioid-related adverse events. These techniques may also allow for better breathing mechanics and reduce atelectasis, but more studies are needed to compare these outcomes. Interestingly, patients in our study who received TEA actually had increased LOS when compared to patients who received GA alone. Zeltsman et al reported similar results in patients undergoing minimally invasive lobectomy who received epidural analgesia as compared to systemic analgesia [29]. Our study showed that patients who received TEA had more comorbidities (14.9% with ASA 4 status) and it is possible that our risk-adjustment was incomplete. In addition, it is possible that TEA was chosen as an analgesic technique for at-risk patients with predicted difficult surgery. It is also possible that other factors related to TEA will be associated with longer LOS. For example, the need in some centers for a dedicated pain service to monitor the patient daily, remove the epidural catheter, and monitor the patient's neurologic status for several hours after catheter removal.

Our study has several limitations. First, the NSQIP database does not differentiate between various regional techniques such as intercostal nerve block, ESP block, paravertebral block, or "single-shot" regional block vs continuous nerve block via catheter. It does not specify if intercostal nerve block administered by the surgeon is classified as regional and if liposomal bupivacaine was used. Second, ACS-NSQIP does not have certain clinically important data such as variables for pulmonary function such as FEV1, surgical complexity and patient frailty, which may have confounded outcomes [30]. Third, there could have been selection bias in which patients received various anesthetic techniques and this may not have been fully controlled for in our modeling. Also, patient outcomes are associated with volume of cases performed at a surgical center and this data is not available in NSQIP database. Fourth, our outcome definition for PPCs did not include some important outcomes such as atelectasis and persistent air leak as these are not available in NSQIP database. Lastly, additional factors that affect PPCs such as intra-operative ventilatory settings were not available in the database, hence were not included in our analysis.

## Conclusion

In summary, our analysis suggests that the addition of regional or local anesthesia to GA is associated with reduced LOS after VATS lobectomy. Patients who received TEA had longer LOS and it possible that at-risk patients with predicted difficult surgery were chosen to receive a thoracic epidural. However, the use of these techniques was not associated with lower PPCs. Future randomized controlled trials could help to elucidate the best anesthetic technique to reduce pain and associated complications.

## Supporting information

**S1 Table. Year of operation by anesthesia technique.**
(DOCX)

**S2 Table. Patient characteristics by group.**
(DOCX)

**S3 Table. ASA physical status classification by anesthesia technique.**
(DOCX)

**S4 Table. Results from linear regression models examining the differences in probability of morbidity/mortality across anesthesia techniques.**
(DOCX)

## Author Contributions

**Conceptualization:** Priyanka Singla, Michael Mazzeffi.

**Data curation:** Brian Brenner, Nabil Elkassabany, Linda W. Martin, Phillip Carrott, Christopher Scott.

**Formal analysis:** Priyanka Singla, Siny Tsang, Michael Mazzeffi.

**Methodology:** Priyanka Singla, Siny Tsang, Michael Mazzeffi.

**Supervision:** Michael Mazzeffi.

**Validation:** Priyanka Singla, Siny Tsang.

**Writing – original draft:** Priyanka Singla.

**Writing – review & editing:** Priyanka Singla, Brian Brenner, Siny Tsang, Nabil Elkassabany, Linda W. Martin, Phillip Carrott, Christopher Scott, Michael Mazzeffi.

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
