## [Decision Letter · Decision Letter 0]

11 Jun 2024

PONE-D-24-12084Anesthetic technique and postoperative pulmonary complications (PPC) after Video Assisted Thoracic (VATS) lobectomy : A retrospective observational cohort studyPLOS ONE

Dear Dr.  Singla,

Thank you for submitting your manuscript to PLOS ONE. After careful consideration, we feel that it has merit but does not fully meet PLOS ONE’s publication criteria as it currently stands. Therefore, we invite you to submit a revised version of the manuscript that addresses the points raised during the review process.

**ACADEMIC EDITOR: **please carefully assess all the reviewers comments

We look forward to receiving your revised manuscript.

Kind regards,

Silvia Fiorelli

Academic Editor

PLOS ONE

Journal Requirements:

3. In the online submission form, you indicated that "Data is available at request from ACS NSQIP database (https://www.facs.org/quality-programs/data-and-registries/acs-nsqip/participant-use-data-file/)"

Reviewers' comments:

Reviewer's Responses to Questions

**Comments to the Author**

1. Is the manuscript technically sound, and do the data support the conclusions?

Reviewer #1: Partly

Reviewer #2: Yes

Reviewer #3: Partly

2. Has the statistical analysis been performed appropriately and rigorously? 

Reviewer #1: I Don't Know

Reviewer #2: Yes

Reviewer #3: I Don't Know

3. Have the authors made all data underlying the findings in their manuscript fully available?

Reviewer #1: Yes

Reviewer #2: Yes

Reviewer #3: Yes

4. Is the manuscript presented in an intelligible fashion and written in standard English?

Reviewer #1: Yes

Reviewer #2: Yes

Reviewer #3: Yes

5. Review Comments to the Author

Reviewer #1: Thank you for letting me review your paper. I appreciate the effort to collect and interpret this large amount of data. I have some questions and concerns. You claim that there is a significant difference in the lenght of stay between the different groups while there is no difference in the risk of having a postoperative complication defined as you have. My main concern about your conclusions of the results are:

1. During the years of inclusion the practice has changed a lot and you point out that TEA becomes less common while regional anestesia becomes a lot more used. You include year of surgery as a covariate in the regression model and there is no significance to what year sugery was done but at the same time the differences in the practice is very large as regional anestesia more than doubled during the 5 year period. This make me question if the impact of time really is insignificant as you interpret it. Maybe I am not good enough on regression models but maybe you could comment on that?

2. Some covariates in the regression analysis is hard to understand why they are included and some could have been included such as pulmonary function FEV1 and DLCO, but maybe they were not available in your register.

3. You mention in your discussion that prolonged air-leak is not part of the composite complication endpoint. That would have been interresting as that is the main reason for prolonged LOS in my experience.

4. The impact of different operating centers are not commented on. As 658 hospitals are contributing to the data there are most likely a big difference in surgical volumes between hospitals which could affect the results. As TEA becomes less common and regional anestesia goes in the oppsite direction you could speculate that a small volume center could be less likely to adopt new routines and in that way TEA might be a confounding factor.

Kind regards

Jesper Andreasson

Reviewer #2: This retrospective cohort study includes a large number of patients and hospitals. The authors analyzed the incidence of post-operative pulmonary complications (PPC) and length of stay (LOS) after lobectomy in video-assisted thoracic surgery (VATS) according to the analgesic technique—epidural, regional, local anesthesia, or none—associated with general anesthesia. They found that between 2017 and 2021, among 15,084 patients from 658 hospitals, 3.5% to 5.2% exhibited a PPC, with a LOS between 3 and 4 days. The subject is interesting as PPC remains a major complication after pulmonary lobectomy. The number of patients is significant, and the data are relatively complete on key elements for analysis. The manuscript is well-written, though a few elements need clarification for the readers as described below:

About PPC criteria: What is the duration of the study period? Are all reintubations within 30 days collected? What criteria are used for pneumonia? Is the diagnosis consistent across different hospitals and based on robust criteria?

Is it pertinent to analyze the data for thoracic epidural analgesia (TEA) in 2021? This data comes from such a small portion of the population (2.8%) that it should at least be mentioned in the discussion.

The distribution of analgesia techniques changed during the study period. Are the hospitals the same throughout the period, or is this linked to a center effect by including data in 2021 from new centers with different practices?

PPC results: Please add the PPC incidence for GA + regional anesthesia for comprehensive information for the reader. Lines 216-218: The results are only summarized at the beginning of the discussion part. There is no description, even a short one, and no reference to the additional material in the results section. However, this is an important aspect of the interpretation of data.

LOS results: The effects on LOS are dramatically increased by the use of Poisson regression. It would be useful for the reader to see absolute values of LOS in the different groups with comparisons before regression. The relevance of the LOS difference could be better considered. Line 193: the reference to the table is incorrect.

The proportion of ASA 4 patients and operation time are higher in the TEA group. It is possible that TEA was chosen for specific at-risk patients with predicted difficult surgery, or that operation time decreased with increased surgical experience during the study period. Mitigation in the interpretation of the LOS results could be better emphasized in the discussion and conclusion to account for this bias.

Line 221 suggests that inadequate analgesia occurs with some types of analgesic techniques. No data on analgesia are presented here, and non-significant results could be due to similar pain scores with different techniques if well-conducted.

Predicted morbidity and mortality scores appear in the statistical analysis method and additional results but are not described or interpreted in the main document, or I did not notice them. Are they still useful to the article, or are they outside the focus of this article?

The typos in the text on lines 184-186 and 82-83 are not consistent.

Please review these points to enhance the clarity and accuracy of your manuscript

Reviewer #3: The authors present a sound written large database analysis on the association of anesthetic techniques with post surgical pulmonary complications. The paper is well written and the discussion In the discussion, conclusions are drawn that are based on the data. like all retrospective work, there are limitations to the informative value. The authors found no correlation between the type of anaesthesia procedure and respiratory complications. however, they do represent an effect on LOS. This difference may be statistically significant, but a difference of less than one day's stay on average appears to be of little clinical relevance. The authors should discuss this point. It would be helpful to show whether there were fewer very long stays in the group with regional procedures.

6. PLOS authors have the option to publish the peer review history of their article (what does this mean?). If published, this will include your full peer review and any attached files.

Reviewer #1: **Yes: **Jesper Andreasson

Reviewer #2: **Yes: **aude carillion

Reviewer #3: No

---

## [Author Response · Author response to Decision Letter 0]

25 Jul 2024

The following files have been uploaded - Response to Reviewers, Revised Manuscript with Track Changes and Manuscript.

Data used in our manuscript is available at NSQIP URL as mentioned on the previous page - https://www.facs.org/quality-programs/data-and-registries/acs-nsqip/participant-use-data-file/

---

## [Decision Letter · Decision Letter 1]

26 Aug 2024

Anesthetic technique and postoperative pulmonary complications (PPC) after Video Assisted Thoracic (VATS) lobectomy : A retrospective observational cohort study

PONE-D-24-12084R1

Dear Dr. Singla,

We’re pleased to inform you that your manuscript has been judged scientifically suitable for publication and will be formally accepted for publication once it meets all outstanding technical requirements.

Kind regards,

Silvia Fiorelli

Academic Editor

PLOS ONE

Additional Editor Comments (optional):

Congratulations to the authors and thanks to the reviewers for the provided suggestions which really helped improve the quality of the manuscript

Reviewers' comments:

Reviewer's Responses to Questions

**Comments to the Author**

1. If the authors have adequately addressed your comments raised in a previous round of review and you feel that this manuscript is now acceptable for publication, you may indicate that here to bypass the “Comments to the Author” section, enter your conflict of interest statement in the “Confidential to Editor” section, and submit your "Accept" recommendation.

Reviewer #1: All comments have been addressed

Reviewer #3: All comments have been addressed

2. Is the manuscript technically sound, and do the data support the conclusions?

Reviewer #1: Yes

Reviewer #3: Yes

3. Has the statistical analysis been performed appropriately and rigorously? 

Reviewer #1: (No Response)

Reviewer #3: I Don't Know

4. Have the authors made all data underlying the findings in their manuscript fully available?

Reviewer #1: Yes

Reviewer #3: Yes

5. Is the manuscript presented in an intelligible fashion and written in standard English?

Reviewer #1: Yes

Reviewer #3: Yes

6. Review Comments to the Author

Reviewer #1: You have adressed the question and comments in my first respons. The use of TEA, at least in Sweden is noow solely used in thoracotomies and therefore a randomised trial would in my opinion not be able to perform as I think the same is true for many other countries.

Reviewer #3: The authors adressed all crucial points and revised the manuscript adequately. I recommend to to accept the manuscript. Congrats.

7. PLOS authors have the option to publish the peer review history of their article (what does this mean?). If published, this will include your full peer review and any attached files.

Reviewer #1: **Yes: **Jesper Andreasson

Reviewer #3: **Yes: **Sven Bercker

---

## [Editor Report · Acceptance letter]

26 Sep 2024

PONE-D-24-12084R1 

PLOS ONE

Dear Dr. Singla, 

I'm pleased to inform you that your manuscript has been deemed suitable for publication in PLOS ONE. Congratulations! Your manuscript is now being handed over to our production team.

Kind regards, 

on behalf of

Dr. Silvia Fiorelli 

Academic Editor

PLOS ONE